# Debonding of Thin Bonded Rubberised Fibre-Reinforced Cement-Based Repairs under Monotonic Loading: Experimental and Numerical Investigation

**DOI:** 10.3390/ma15113886

**Published:** 2022-05-30

**Authors:** Syed Asad Ali Gillani, Shaban Shahzad, Wasim Abbass, Safeer Abbas, Ahmed Toumi, Anaclet Turatsinze, Abdeliazim Mustafa Mohamed, Mohamed Mahmoud Sayed

**Affiliations:** 1Laboratoire Matériaux et Durabilité des Constructions, Université de Toulouse, Institut National des Sciences Appliquées de Toulouse, UPS Génie Civil, 135 Avenue de Rangueil, CEDEX 04, 31077 Toulouse, France; asadgillani@uet.edu.pk (S.A.A.G.); shahzad@insa-toulouse.fr (S.S.); toumi@insa-toulouse.fr (A.T.); anaclet@insa-toulouse.fr (A.T.); 2Civil Engineering Department, University of Engineering and Technology, Lahore 54890, Pakistan; safeer.abbas@uet.edu.pk; 3Department of Civil Engineering, College of Engineering, Prince Sattam Bin Abdulaziz University, Al-Kharj 16273, Saudi Arabia; a.bilal@psau.edu.sa; 4Building & Construction Technology Department, Bayan College of Science and Technology, Khartoum 210, Sudan; 5Architectural Engineering, Faculty of Engineering and Technology, Future University in Egypt, New Cairo 11745, Egypt; mohamed.mahmoud@fue.edu.eg

**Keywords:** fibres, rubber particles, thin bonded overlay, debonding, DIC, CAST3M

## Abstract

In this study, the durability of cement-based repairs was observed, especially at the interface of debonding initiation and propagation between the substrate–overlay of thin-bonded cement-based material, using monotonic tests experimentally and numerically. Overlay or repair material (OM) is a cement-based mortar with the addition of metallic fibres (30 kg/m^3^) and rubber particles (30% as a replacement for sand), while the substrate is a plain mortar without any addition, known as control. Direct tension tests were conducted on OM in order to obtain the relationship between residual stress-crack openings (σ-w law). Similarly, tensile tests were conducted on the substrate–overlay interface to draw the relationship between residual stress and opening of the substrate–overlay interface. Three-point monotonic bending tests were performed on the composite beam of the substrate–overlay in order to observe the structural response of the repaired beam. The digital image correlation (DIC) method was utilized to examine the debonding propagation along the interface. Based on the different parameters obtained through the above-mentioned experiments, a three-point bending monotonic test was modelled through finite elements using a software package developed in France called CAST3M. Structural behaviour of repaired beams observed by experimental results and that analysed by numerical simulation are in coherence. It is concluded from the results that the hybrid use of fibres and rubber particles in repaired material provides a synergetic effect by improving its strain capacity, restricting crack openings by the transfer of stress from the crack. This enhances the durability of repair by controlling propagation of the interface debonding.

## 1. Introduction

Concrete has been utilised abundantly in the construction sector over recent decades, and with the passage of time, a reduction in the load-retaining capacity of existing infrastructures has been observed. In order to rehabilitate damaged concrete structures, different techniques have been used. Among different rehabilitation techniques, a unique approach to reinstate the performance of a degraded structure is thin bonded cement-based overlay. This overlay technique is used to replace the decaying concrete, to provide smoothness to the damaged part of structure and to enhance the load carrying capacity through increased thickness, which also provides an extra margin for protecting it against corrosion [1]. Such a technique proves to be very efficient, specifically for larger surfaces of concrete such as pavements [2,3].

However, the durability of these overlays can be limited due to cracking of the repaired part, followed by the interface’s delamination from the substrate [3,4]. This issue has already been well-reported in previous studies [1,5,6]. According to some previous research [1,3,4], mechanical loadings and differential shrinkage are the major causes for the delamination between overlay and substrate. The delamination normally begins from edges, cracks and joints in all mechanisms.

On the basis of previous literature, one can say that the long-term sustainability of the materials used for repair, and bonding between two layers, are merely influenced by the durability characteristics of thin bonded overlays. As for sustainable materials, reliable option to improve the durability properties of the repair system is to use rubber aggregates and steel fibres collectively [7,8,9,10]. The inclusion of rubber particles obtained by grinding scrape tyres in repair composites enhances their strain capacities [9,11,12,13], and fibres limit the crack opening, which assists in delaying debonding initiation and restricting the interfacial delamination to greater extent. Moreover, positive synergetic effects (enhancement in the strain capacity of material and in post-cracking residual tensile strength) were found by the collective use of rubber particles and fibres in mortar [7,8,10,12]. Due to these positive synergetic effects, use of rubber particles and fibre in cement-based overlay is most often adopted.

Several studies have been performed to analyse the crack and propagation of delamination in cement-based overlays under different kinds of mechanical or thermal loadings. Gillani et al. [14] studied the generation and movement of crack and delamination of the overlays under fatigue loading. They found that the addition of metallic fibres and rubberised particles help to control the debonding by restraining the crack (with addition of fibres), as well as by improving the strain capacity (with addition of rubber aggregates). Mateos et al. [15] reported the mechanical behaviour of the asphalt–concrete interface in a bonded concrete overlay of asphalt pavements (BCOA). Cylindrical specimens were used under various conditions such as wet and dry, and temperature ranges between 5 and 40 °C. The results indicate that the strength of the concrete–asphalt interface is strongly linked with the asphalt. Moreover, interface significantly softened under wet conditions, indicating that water is the decisive factor responsible for the failure of BCOA. However, one can conclude that concrete has not developed a good bond with asphalt. A. Toumi et al. [12] conducted experimental and analytical study on delamination of a thin rubberised and fibre-reinforced mortar repair. In this study, substrates of cement-based material (100 mm-thick) and an overlay with cement-based composites containing fibres and/or rubber particles (40 mm thick) were used. The study was conducted under a three-point bending test using a monotonic sequence of loading. They found that the addition of fibres in repair material helps to delay the debonding phenomena, and the inclusion of the rubber particles improves the strain capacity of the material, resulting in controlling of the debonding compared to the repair material without rubber aggregates. Studies were conducted to analyse the debonding of substrate and fibre-reinforced mortar (FRM) overlay by Q.T. Tran et al. [4]. A substrate in the form of a hollow metal beam was used in this study. The test was conducted under static three-point bending conditions. Experimental results were compared with the results obtained through the model. The finite element model (FEM) was based on the discrete crack model, which helps to model the crack and debonding propagation efficiently. The numerical results show that the developed model is an effective system to forecast the crack opening and debonding propagation.

A study on the behaviour of fibre-reinforced concrete (overlay) over the asphalt (substrate) was conducted by Isla et al. [16]. Bending tests were carried out on specimens of size 100 × 100 × 400 mm with centre-to-centre distance of supports of 350 mm, and a thickness of 50 mm per layer for concrete overlay and substrate of asphalt. Isla et al. [16] reported that inclusion of fibres significantly improved the residual capacity of flexural member and composite beams as well. Hasani et al. [17] has also reported that the overlay of fibre-reinforced concrete also improves the mechanical and durability-related properties. Moreover, it was found that the compressive strengths, flexural strength, residual strength, and ductility of the FRC overlay material was improved. However, the modulus of elasticity was reduced.

On the basis of previous studies, it can be concluded that the bond between the overlay and substrate and characteristics of the repair material significantly affect the durability-related properties of thin overlay systems. Moreover, the debonding mechanism between the overlay and substrate initiates when the crack reaches at interface. In this regard, the hybrid use of fibre and rubber aggregates appears to be a viable option to improve the durability characteristics of repair system. The current research was planned to investigate the flexural behaviour of composite beams under monotonic load. The evolution of crack opening, deflection and debonding length was evaluated to study the potential of rubberized fibre-reinforced composite material for possible utilization as a repair material in cementitious overlays. To ensure a good bond between the repair material and substrate, the sandblasted substrate surfaces were used as per previous research findings [18]. To analyse the crack’s evolution and delamination at the interface, the DIC method was used. The flexural tests were modelled using the finite element approach based on a discrete crack model to forecast the crack propagation and debonding mechanism under monotonic loading.

## 2. Materials

Cementitious mortar including rubber particles with the addition of fibres was used as a repair material in this study. Portland cement (CEM I 52.5R) in conformity with EN197-1:2011 [19] and natural sand (0–4 mm) were used. The chemical composition and physical properties of the Portland cement are shown in Table 1. Similar findings have also been reported in other research study [20]. Master Glenium 27, a modified polycarboxylic ether polymer-based superplasticizer, and Rheomac were used as superplasticizer and viscosity-modifying agent (VMA), respectively. The rubber particles were used as a partial replacement of sand in the same volumetric unit. The rubber aggregates were produced through grinding of scrap tyres. Rubber particles’ specific gravity was 1.2, which is much less than sand, i.e., 2.7. Gradation curves of rubber aggregates and sand show slightly different particle size distribution for both materials, but in both cases, the maximum particle size is limited to 4 mm, as can be visualized in Figure 1. Fibraflex Saint-Gobain [21] provided the fibres, shown in Figure 2. The length of these fibres is 30 mm, which also meets the criteria for productive bridging properties, i.e., the maximum aggregate particle size should be equal or less than half of the length of fibre [22]. These fibres develop an excellent bond with cementitious composites because of the rough and large surface area. Properties of these amorphous metallic fibres are provided in Table 2 [3]. Similar mixture proportions from another study were adopted for the current work [4], [11], and are presented in Table 3. In view of previously conducted research [11], the maximum dosage of fibre of 30 kg/m^3^ was investigated and the partial replacement of sand aggregate with rubber particles was 30%.

The mixtures were designated with R and F for rubber aggregates and fibres, respectively, for referencing of different mixtures. For instance, M0R30F is designated mixture refer as follows:

M represents mortar, 0R shows 0% rubber particle, and 30F denotes mixture with 30 kg/m^3^ of fibres.

By keeping the same water-to-cement ratio (w/c), the quantity of super-plasticizer was changed to keep the same workability with a slump of 10 ± 2 mm. In fibre-reinforced and/or rubberized mortars, the quantity of super-plasticizer is required to be increased because of the decrease in the workability of mortar with the addition of fibres [17]. Additionally, it was observed that air content increases to 65% by the inclusion of rubber aggregates in mortar, as projected in the literature [23]. Rubber aggregates are lightweight and water-repellent which makes them very susceptible to segregation. The role of the viscosity agent is to avoid this detrimental phenomenon.

## 3. Mechanical Characterization

### 3.1. Compressive Tests

Tests for compressive strength were carried out in accordance with European standard, NF EN 12390-3 [24]. Specimens used for compressive tests were in accordance with EN 12390-1 [25]. Cylindrical specimens with a 110 mm diameter and height of 220 mm were prepared.

### 3.2. Modulus of Elasticity Tests

Elastic modulus tests were carried out on studied mixture composites, using the same specimen size as in Section 3.1 for each material. These tests were conducted by following the standard NF EN 12390-13 [26]. A cage with three attached extensometers at an equal angle from each other was used, as shown in Figure 3. The stress–strain relationship was plotted using average deformation in a longitudinal direction with mounted extensometers.

### 3.3. Direct Tensile Tests

Prismatic notched specimens with a size of 100 × 100 × 200 mm and reduced cross sections of 50 × 50 mm, as shown in Figure 4, were prepared for direct tensile testing. These tests were conducted to assess the tensile properties and stress–crack relationships for various composites. These will be the input factors for the finite element model. These tests were conducted as per the RILEM recommendation [27]. An MTS press was used for conducting the test and CMOD was recorded by using a COD clip, as shown in Figure 5. One can analyse the capacity of the deformation linked with peak load and residual tensile strength beyond the peak through these tensile tests. A loading speed of 5 μm/min was adopted in the start of the test till the CMOD reaches 0.1 mm, and then the loading rate was increased to 100 μm/min until failure of the specimen.

## 4. Bending Monotonic Test

### 4.1. Specimen for the Monotonic Test

The composite samples were made of a thin repair layer applied on top of the substrate, which mimics the repaired beam. Cementitious substrates without rubber aggregates and fibre-reinforcement (M0R0F) were prepared to have a real application. The size of prismatic substrate was 100 mm × 100 mm × 500 mm. These substrate bases were placed in a control environment of 20 °C and relative humidity (RH) of 98% for curing purposes. Based on the results from previous studies [18,28,29,30,31,32,33,34], it is observed that the surface preparation of the substrate has an influential role on the performance of the repair as far as durability is concerned. So, substrates prepared by the sandblasting techniques were used in experiments. In reference to previous studies [34,35], a 40 mm-thick layer was used as the repair. So, a repair layer was cast on top of the 100 mm sandblasted substrates. A schematic diagram of the beam with a repair layer under three-point bending monotonic testing is shown in Figure 6. To predetermine the location of the crack, the repair layer was notched during the specimen’s casting at the mid-span. These beams were placed under ambient conditions (20 °C and RH of 98%) for 28 days.

### 4.2. Testing Procedure

A three-point bending monotonic test was performed on the specimen to analyse the behaviour of the overlay–substrate under flexure. The schematic testing setup can be seen in Figure 6. CMOD was measured by using a COD sensor. A loading speed of 0.05 mm/min was adopted at the start of the test until the CMOD reached 0.1 mm, and then the loading speed was increased to 0.2 mm/min until failure of the specimen (when resisting load is equal to around zero). The LVDT sensor was used to monitor the vertical deflection of the composite specimens at the middle. For monitoring the interface delamination and crack propagation, a digital DIC technique was used. Under mechanical loading, crack initiated from the tip of the notch in the overlay, which eventually caused the delamination when it approached the interface. The main objective of the monotonic tests is to monitor the following parameters:The opening of the notch (CMOD) with the application of force.The deflection with force.The load at which the crack approaches the interface location with DIC.

### 4.3. Digital Image Correlation Technique

The DIC method was developed by researchers from University of California in the late 19th century [36,37,38,39]. DIC is a visual and non-contact measurement technique that is used for monitoring of surface displacements of an object under investigation by image registration techniques for accurate measurement of changes in images taken in series with test proceeding. Strain on the surface of the object is calculated using the displacements. Random speckles are made on the white painted surface of the object prior to the initiation of the test to obtain the most effective results [40].

Two images were taken from two cameras within the same period of time using the 3D DIC technique. The system must be calibrated prior to the test. After calibration, these results can be used correlate the images for the determination of the deflection and strain of the object under investigation [41]. For 3D image correlation, preparation of the specimen is necessary, as shown in Figure 7. The complete testing layout for DIC can be seen in Figure 8.

Three-point monotonic bending tests with DIC technique were conducted for all repairs to examine the pattern of the crack and to evaluate the load where the crack approaches the interface and debonding starts. The software Vic-3D, [42] was used for image processing. The generated displacement and strain on the surface of the object can be analysed through post-treatment of images taken during the bending test. The complete cracking pattern is shown by the post-treatment. An artificial extensometer is used to find the value of the load at which the crack propagates to the interface. The placement of an artificial extensometer on the surface is shown in Figure 9. The DIC method has the ability to carry out reverse analysis of obtained strains through post-processing. So, an artificial extensometer can be placed at an actual crack location after locating the crack path to detect the load where the crack approaches the interface. When the crack passes this extensometer, an abrupt variation in D1 value is observed (Figure 10). The D1 is an extension in the artificial extensometer that develops due to crack opening. At the location where there is an abrupt change in D1, the corresponding load value represents the propagation of the crack to the interface and initiation of the interface debonding.

The loads required to start the delamination of various specimens with unique repair mixtures are provided in Table 4. For M0R0F and M30R0F repair mortars, the average force for debonding initiation is comparatively less for M0R30F and M30R30F. These fibres provide bridging properties throughout the crack path and reinstate the crack opening. For M30R0F repairs, the load needed to start the delamination is also improved in comparison to M0R0F mortar repair. The inclusion of rubber particles increases the capacity of strain in the material and helps in delaying crack initiation. Additionally, the delamination initiation load is notably enhanced for M30R30F. The increase in the load value is due to the synergetic effect induced by the collective use of rubber particles and fibres.

## 5. Numerical Modelling

Various discrete crack models were developed for modelling and analysing the behaviour of normal concrete and fibre-reinforced concrete under monotonic loading. Petersson [43] used fictitious crack models based on fracture mechanics, enabling the prediction of the growth of crack and fracture zones in normal concrete or composite concrete. In the developed model, cracks in the overlay and interface debonding were propagated according to Mode I of fracture mechanics. The initiation as well as the propagation of the crack in the overlay followed the pre-damaged path, or along the zone of minimum strength (at the location of notch).

### 5.1. Mesh Size of Composite Beam

As per the symmetrical system, only one half of the beam was modelled for the optimization of the simulation by the FEM package. Figure 11 shows the model of half of the composite beam in the software. Triangular and rectangular elements were made with three and four nodes, respectively, to have an optimised mesh size. To obtain accurate and optimised results from model, Tran [34] analysed the behaviour of a composite beam by altering the size of the mesh to obtain stabilised results. He observed that while using inter-node distances less than 1mm, the effect of mesh size on the results became insignificant. In this study, a node-to-node distance of 1 mm was selected.

### 5.2. Cracking/Delamination Modelling Theory

Three-point bending tests under monotonic loading on the repaired beams as described in Section 3 were modelled using the discrete crack model mentioned earlier. To model mechanical response of fibre-rubberised mortar, material characteristics were described for the overlay and interface in Section 6. The stress–crack-opening relationships for overlay materials are given in Equations (1)–(4) as well as for interface in Equation (5).

CAST3M, developed in France by the Atomic Energy Commission, was used for calculation purposes in the finite element method (FEM). To control the propagation of delamination (debonding) or cracks, the stress premier node after the tip of crack of debonding, as proposed in previous studies [4,44], was considered. This type of technique prevents controlling the propagation by the condition or state of stress calculated at the interface or at the crack tip, where the stress singularity was predicted based on the strength theory analysis. So, the tip of the crack or delamination initiation node is advanced to the next node when the state of the stress at the commanding node does not satisfy a stability criterion (depending on the material tensile strength or on the one of the interfaces). The interlocking is considered by closing forces applied between the nodes in front of each other along the crack or along the de-bonded area. Debonding is started through tension perpendicular to the interface with the use of cementitious materials [45]. So, in the studied model, the loads were found through the residual stress–debonding opening/crack (σ-w) laws. By using a hypothesis of plain stress, trilateral and quadrilateral elements were modelled in 2D. At the time of calculation, the interlocking closure load of each particular pair of nodes facing each other were recalculated based on the crack or debonding openings each time a node was freed, to advance the crack or debonding. Residual closing forces were precisely fitted by an iterative approach until fracture widths or debonding widths were stable, according to the stated criterion. Then, for comparison with the propagation criterion, the same method is used on the next controlling node.

Debonding or cracks moved to the next node, if *σ_t_* > *R_t_* or *σ_ti_* > *R_ti_*, respectively, where σ_t_ is the tensile stress, R_t_ is the tensile strength of the repair materials and σ_ti_ and R_ti_ are the tensile stress and tensile strength of the substrate–repair interface, respectively.At the reach of stable state, and next step of increased loading was imposed to restart the propagation if the above-mentioned criterion is not met.

## 6. Results and Discussions

### 6.1. Modulus of Elasticity and Compressive Strength

Compressive strength and modulus of elasticity for all compositions are provided in Table 4. For mortars with rubber aggregates, notable depreciation in compressive strength is found. These results are in line with past studies on the effect of rubber particles partly replacing sand in cement-based materials [11,46]. The compressive strength of composites is not remarkably modified by the addition of fibre-reinforcement. Not only the low stiffness of rubberised aggregates, but also the high porosity and weak ITZ between cementitious and rubber particles had deleterious effects on the mechanical properties of mortar [47].

Similarly, a remarkable reduction in E values of the material was seen due to the inclusion of rubber particles, which is one of the same results found in past studies in this area [11,46]. Low stiffness and an increase in the porosity due to the addition of rubber particles are the main factors for the reduction in the E values. Like the results for compressive strength, the addition of metallic fibres has no effect or a very minute influence on E-values, as provided in the information in Table 5.

### 6.2. Tension Test for Material Used as Repair

Results of the direct tensile experiment for various repair materials are presented in Table 5. The reduction in tensile strength for M30R0F as well as the enhancement in the strain capacity is also observed (around 1.5 times increase in strain capacity vs. that of control mortar), as shown in Figure 12 and Figure 13. Poor formation of ITZ and an increase in the porosity of the composite due to the addition of rubber particles are the main factors for the reduction in tensile strength. Even with the low strength in tension, the deformation at maximum load is higher for material containing rubber particles than the control one, because rubber particles have the capability to withstand failure deformation after peak [48]. From Figure 12, it can be analysed that for M0R30F, residual strength in tension is notably improved. A 3.5 times increase in the strain capacity is observed in M30R30F as well as enhancement in the residual strength in the post-cracking zone compared to the control. Figure 14 shows the experimental results obtained for various materials used for repair. By using best-fit curves of the results obtained through experiments, the following equations were finalised.

For M0R0F
(1)$$\sigma_{t}=R_{t}\times EXP\left(-3\frac{w}{w_{l}}\right)$$

For M0R30F
(2)$$\sigma_{t}=R_{t}\times EXP\left(-2.5\;\frac{w}{w_{l}}\right)$$

For M30R0F
(3)$$\sigma_{t}=R_{t}\times EXP\left(-1.5\;\frac{w}{w_{l}}\right)$$

For M30R30F
(4)$$\sigma_{t}=R_{t}\times EXP\left(-1.3\frac{w}{w_{l}}\right)$$
where “*σ_t_*” is the residual strength in tension, “*R_t_*” is the strength in tension, “*w*” is evolution of the crack opening during loading and “*w_l_*” is the controlling value of the crack opening, after which residual strength in tension becomes negligible or zero. “*R_t_*” and “*w_l_*” for the various materials are summarised in Table 6.

### 6.3. Tension Test for Overlay/Repair–Substrate Interface

The principle of this test was as same as explained in Section 3.3. The objective was to have the analysis of the residual stress-delamination opening and tensile strength analytical relation for the substrate–overlay interface. The tested samples consist of old substrate and new overlay and were notched before testing at the interface on the ends facing each other. The model curve is also shown in Figure 15 based on the exponential model Equation (5).
(5)$$\sigma_{ti}=R_{ti}\times EXP\left(-4\frac{w}{w_{li}}\right)$$
where, “*σ_ti_*” is residual strength in tension, “*R_ti_*” is the interface tensile strength (1.00 MPa), “*w*” is the opening of debonding and “*w_li_*” is the controlling value of debonding opening beyond which strength in tension becomes negligible (0.1250 mm).

### 6.4. Relationship between Force and Opening of Notch

Figure 16 illustrates the relationship between notch opening and the force in the overlay. A comparison between numerically obtained results and experimental ones has been carried out and a good agreement have been observed.

The results obtained from model and the experimental campaign indicate that at any opening of the notch in the overlay, the corresponding load is higher in case of fibre-reinforced mortar compared to the control one. Repair material with fibre-reinforcement limits the notch opening during testing by controlling the opening of the crack.

The addition of rubber particles has no notable effect on the notch opening, as observed from the results. The M0R0F and M30R0F repair materials show an approximately similar response.

### 6.5. Relationship between Force and Deflection

Figure 17 show the relationship between force and deflection. A comparison between simulated and experimental results have been carried, out and an excellent coherence has been observed.

From the results, it is observed that in beams repaired with fibre-reinforced composite, load carrying capacity of the repair material is increased at the corresponding deflection. This is because of the capacity of the fibres to transfer the stress across the crack, which restricts the deflection to a greater extent than in the other repair materials.

### 6.6. Relationship between Force and Debonding

Figure 18 show the relationship between force and debonding propagation at the interface of the composite beams repaired with different materials. In these figures, a comparison between simulated and experimental results has been carried out and an excellent coherence has been noticed.

Figure 19 shows the results obtained from all repair materials in order to provide a good comparison. The test results show that propagation of debonding is more dominant in the repaired material without the fibres (M0R0F and M30R0F). With the addition of fibres in the repaired material (M0R30F and M30R30F), resistance in the debonding along the interface was noticed. This is closely linked with the crack opening in the repair layer. Therefore, the fibre-reinforced repair composites limit crack opening and thus delay the initiation interface debonding and limit propagation.

For M30R30F repair, it is also depicted that for the debonded length, the representing load is higher more than the other ones. For example, to obtain a 20 mm delamination at the interface, 6.5 kN of force is needed by M0R0F repair, 10 kN by M0R30F and 10.7 kN by M30R30F. Similarly, the force needed to initiate the delamination is also higher with M30R30F repair.

Notch opening plays a significant role in the transmission of debonding along the interface, as shown in Figure 20. The M0R0F as a repair material depicts the highest notch opening, and the corresponding debonded length is also the longest of the materials. On the contrary to the previous repair material, the M30R30F repair material restricts the opening of the notch and debonding propagation.

The notch openings of 15, 16, 21 and 22 μm were recorded, at which debonding is initiated in beams repaired with mixed compositions M0R0F, M0R30F, M30R0F and M30R30F, respectively. This indicates that the benefit of the inclusion of rubber aggregates in repair material is not only restricted to crack controlling, but it is also useful to retard the initiation of delamination.

Fibre-reinforcement has no notable effect on the start of delamination. The lengths of delamination are restricted with use of M0R30F and M30R30F as repair materials. Therefore, it is concluded that the repair material reinforced with fibres is not only limited to controlling the crack opening, but also to the debonding.

## 7. Conclusions

In this paper, a detailed experimental and numerical study has been conducted on the structural performance of beams with base and repair under three-point bending monotonic loading. The following conclusions are drawn from the experimental and numerical investigation:The maximum crack restraining was shown by repair material including fibres with or without the inclusion of rubber particles.To initiate the debonding, repair materials that are fibre-reinforced (either with or without rubber particles) required a greater value of load than the repair materials without the inclusion of fibres.M0R0F shows the minimum resistance to initiating interface debonding. However, for M30R0F repaired material, the transmission of delamination is restricted and retarded compared to the control one. The micro-cracking was controlled by rubber particles, which as a result increases and the load for debonding. Moreover, improved strain capacity of the repaired material with the addition of rubber particles also increases the notch opening at which the delamination starts.Debonding transmission at the interface is more significant in the materials without fibres (M0R0F or M30R0F) and is largely controlled in M30R30F repair. It shows the positive synergetic effect by the collective utilisation of rubber particles and fibres under static bending test.The FEM model results show that it accurately predicts the mechanical response of the material under monotonic flexure testing. In particular, it allows the kinetics of crack advancement and of delamination at the interface to be predicted.The developed FEM is an effective technique to predict and analyse the response of the repaired system under mechanical conditions of loading. In particular, it helps to highlight the benefit of incorporating rubber aggregates and fibre-reinforcement and the positive synergetic effect of both.The DIC technique is a suitable tool for the monitoring of debonding propagation along the interface.Finally, as added value to the contribution towards achieving more sustainable repair, the addition of rubber aggregates obtained through grinding of scrap tyres in cementitious materials can also be considered to maintain a clean environment by recycling used tyres and minimizing the use of landfill for residual waste.

## Figures and Tables

**Figure 1 materials-15-03886-f001:**
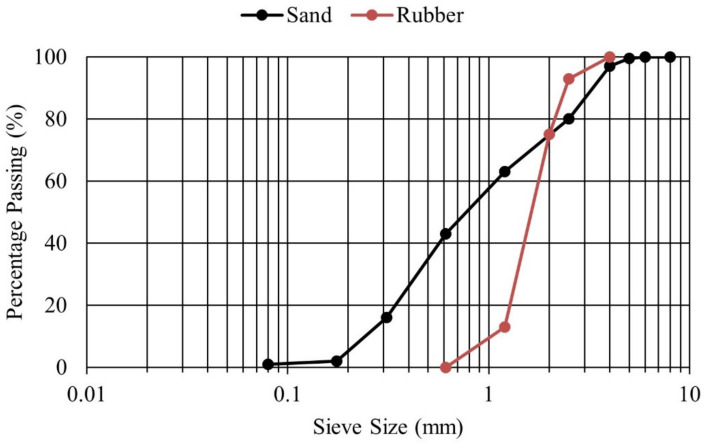
Gradation curve for rubber particles and sand.

**Figure 2 materials-15-03886-f002:**
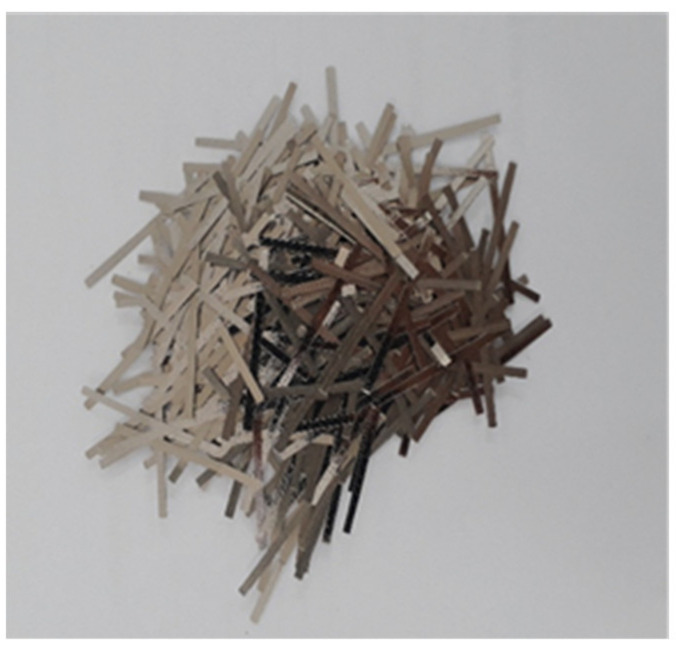
30 mm-long metallic fibres.

**Figure 3 materials-15-03886-f003:**
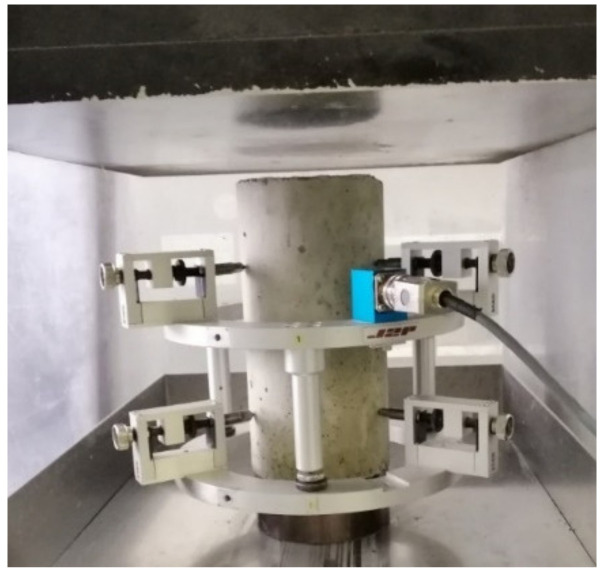
Testing arrangement for modulus of elasticity with cage.

**Figure 4 materials-15-03886-f004:**
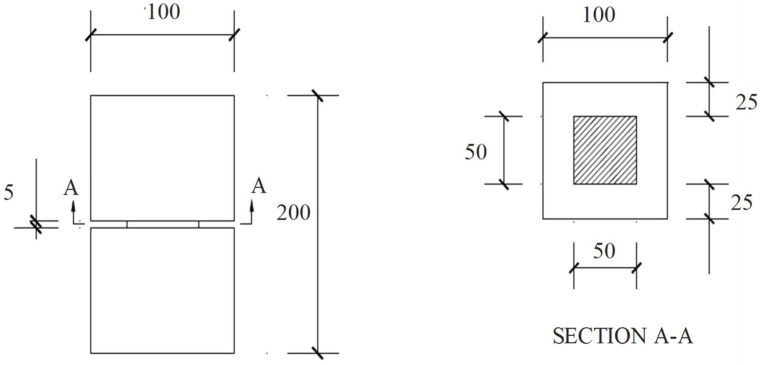
Prismatic notched specimens for direct tensile test (mm).

**Figure 5 materials-15-03886-f005:**
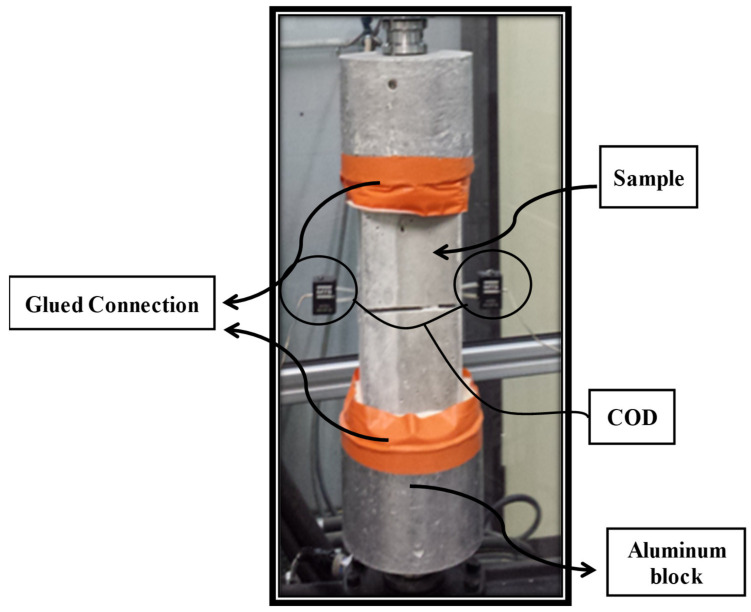
Experimental setup for direct tensile test.

**Figure 6 materials-15-03886-f006:**
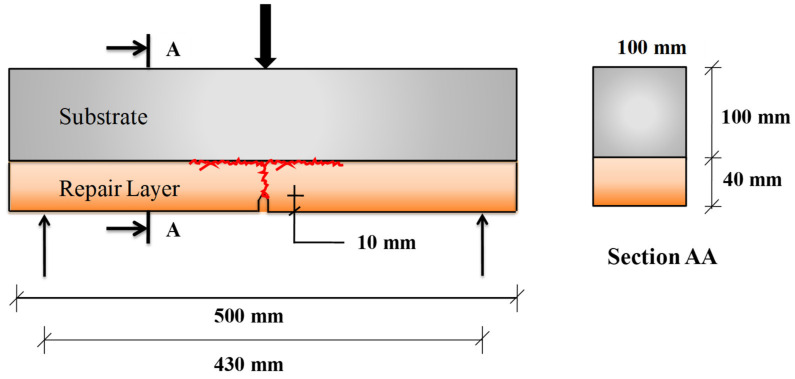
Schematic representation of tested specimen for use in bending test.

**Figure 7 materials-15-03886-f007:**
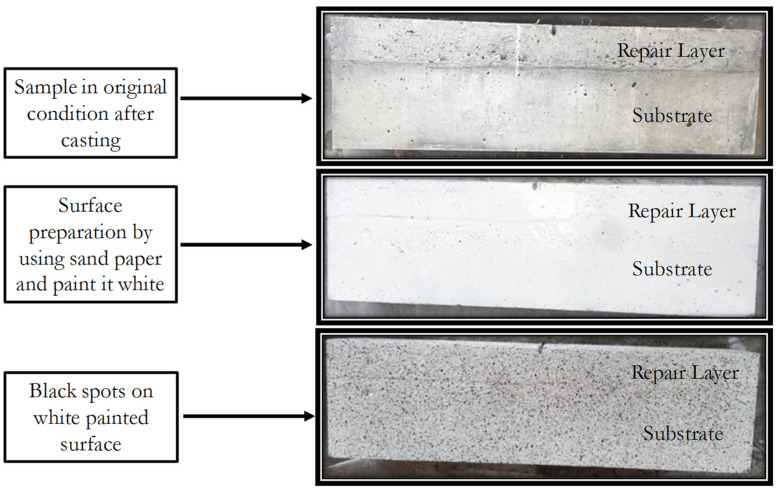
Surface preparation of specimen for the bending test along with DIC technique.

**Figure 8 materials-15-03886-f008:**
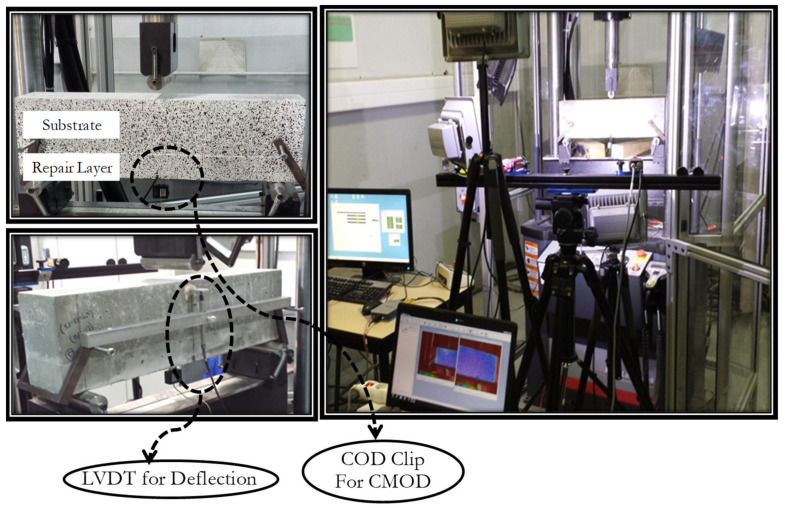
Complete testing layout for three-point bending monotonic test using DIC technique.

**Figure 9 materials-15-03886-f009:**
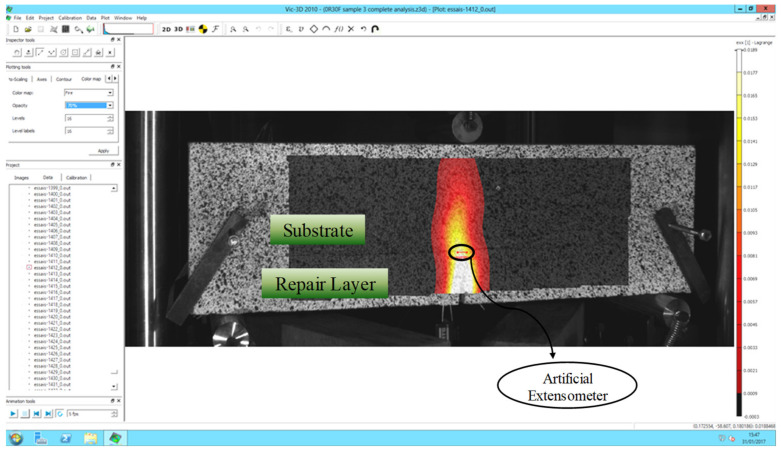
Placement of an artificial extensometer at the interface.

**Figure 10 materials-15-03886-f010:**
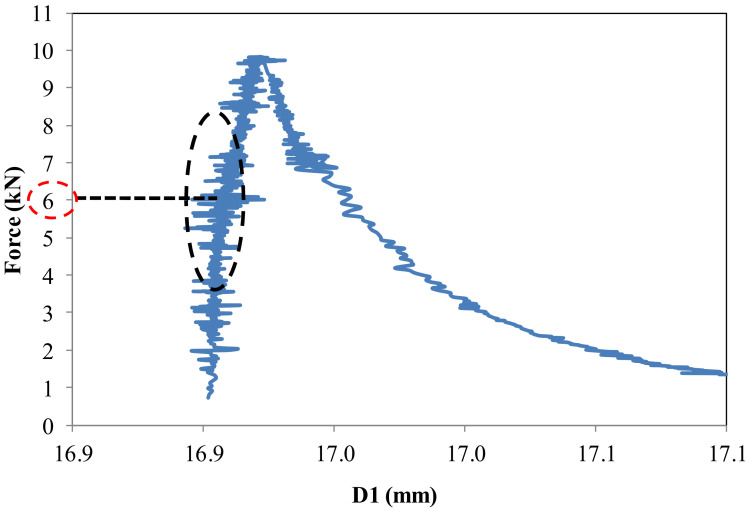
Force versus elongation in artificial extensometer (D1).

**Figure 11 materials-15-03886-f011:**
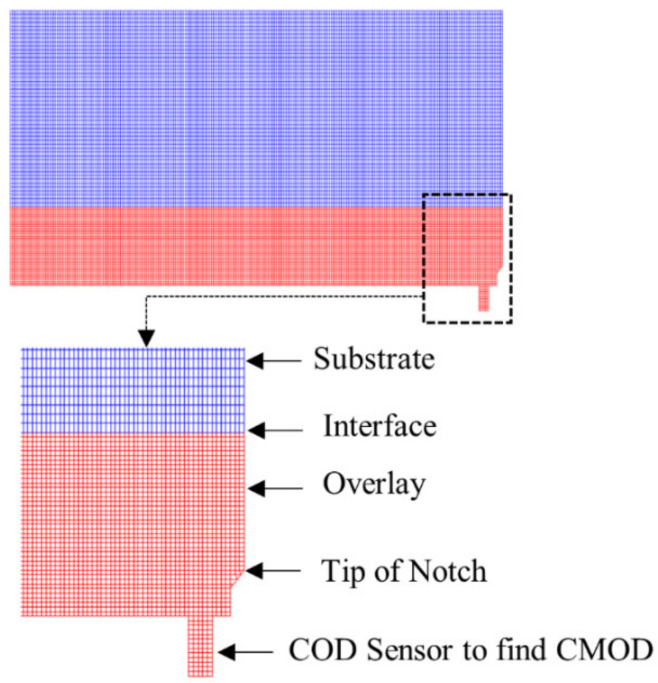
Substrate–overlay composite beam model in FEM software.

**Figure 12 materials-15-03886-f012:**
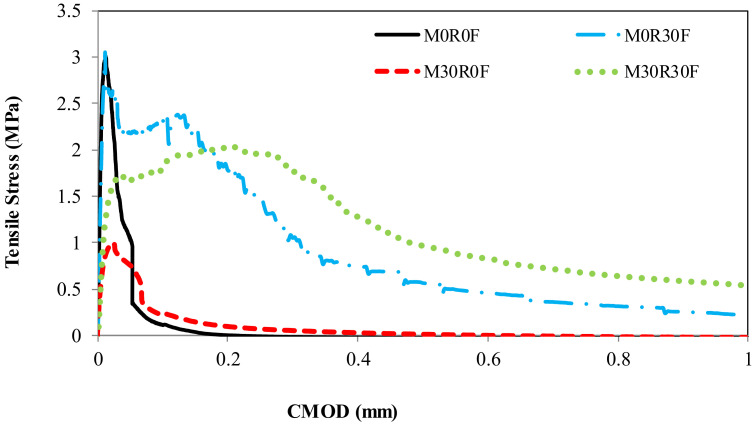
Impact of rubber aggregates and fibres on strain capacity and on residual post-peak strength.

**Figure 13 materials-15-03886-f013:**
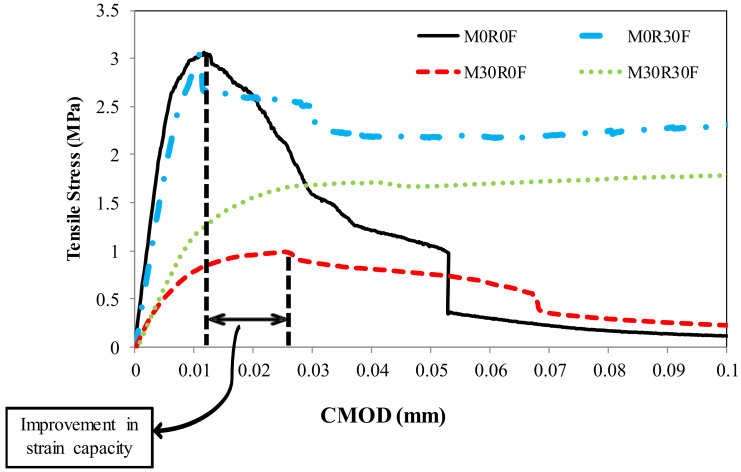
Impact of rubber aggregates and fibres on strain capacity and on residual post-peak strength (enlarged view).

**Figure 14 materials-15-03886-f014:**
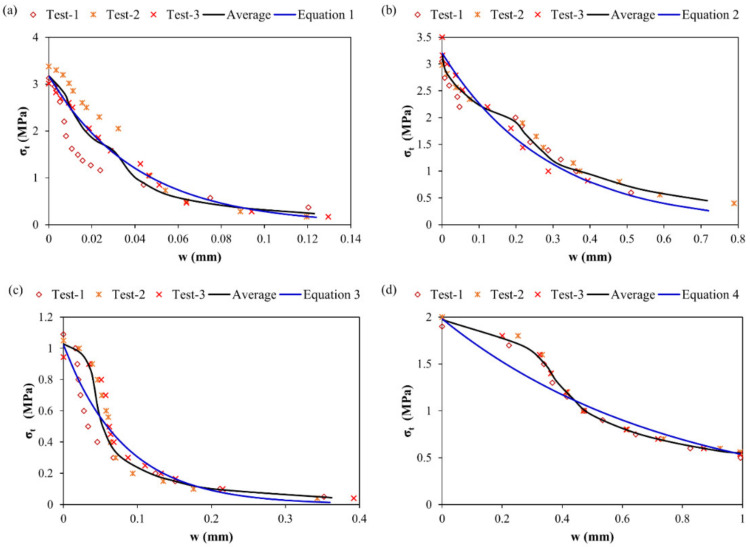
σ-w law for various composites (**a**) plain mortar, M0R0F (**b**) M0R30F (**c**) M30R0F (**d**) M30R30F.

**Figure 15 materials-15-03886-f015:**
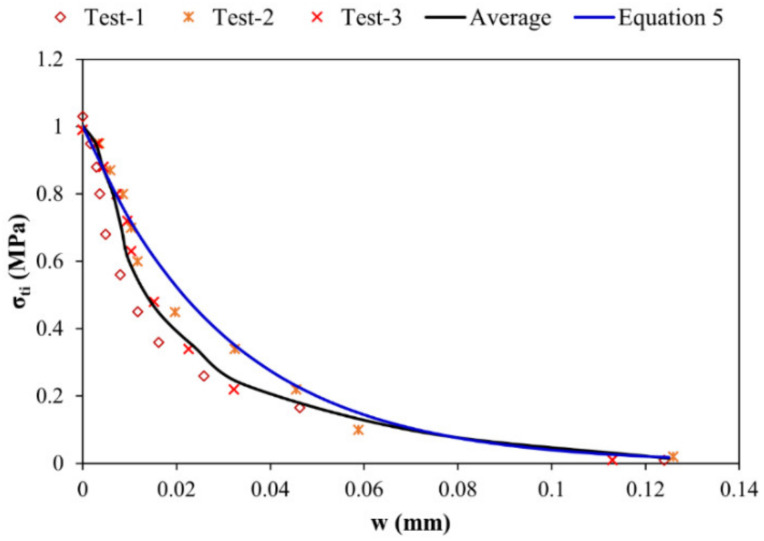
σ-w law for calibrated model and experimental results.

**Figure 16 materials-15-03886-f016:**
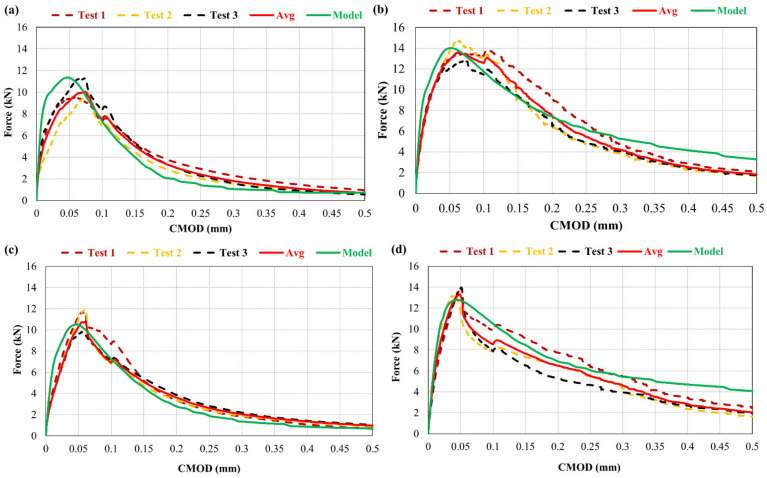
Force vs. opening of notch (CMOD) for (**a**) M0R0F-M0R0F, (**b**) M0R0F-M0R30F, (**c**) M0R0F-M30R0F, and (**d**) M0R0F-M30R30F.

**Figure 17 materials-15-03886-f017:**
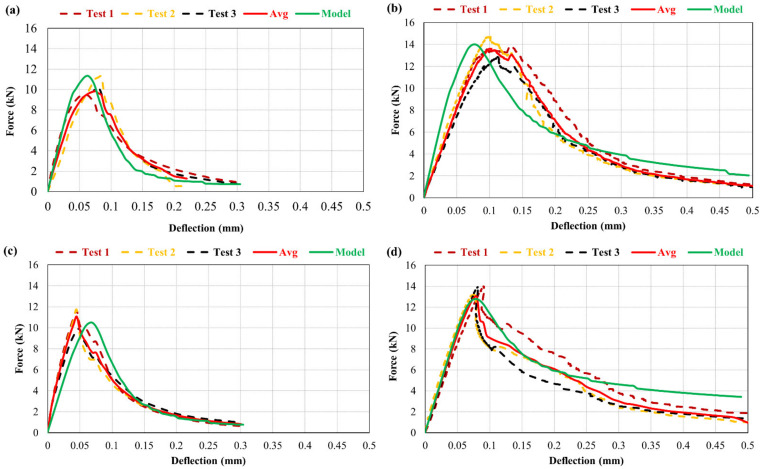
Force vs. deflection in composite beam for; (**a**) M0R0F-M0R0F, (**b**) M0R0F-M0R30F, (**c**) M0R0F-M30R0F, and (**d**) M0R0F-M30R30F.

**Figure 18 materials-15-03886-f018:**
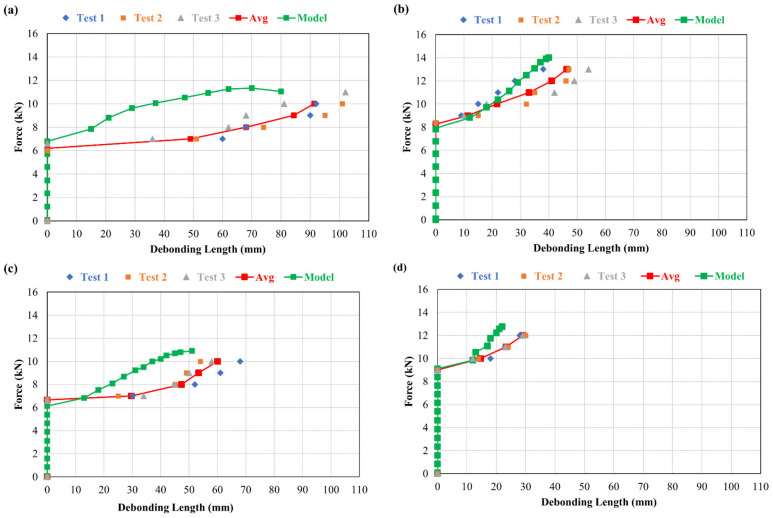
Force vs. debonding at interface in composite beam for; (**a**) M0R0F-M0R0F, (**b**) M0R0F-M0R30F, (**c**) M0R0F-M30R0F, and (**d**) M0R0F-M30R30F.

**Figure 19 materials-15-03886-f019:**
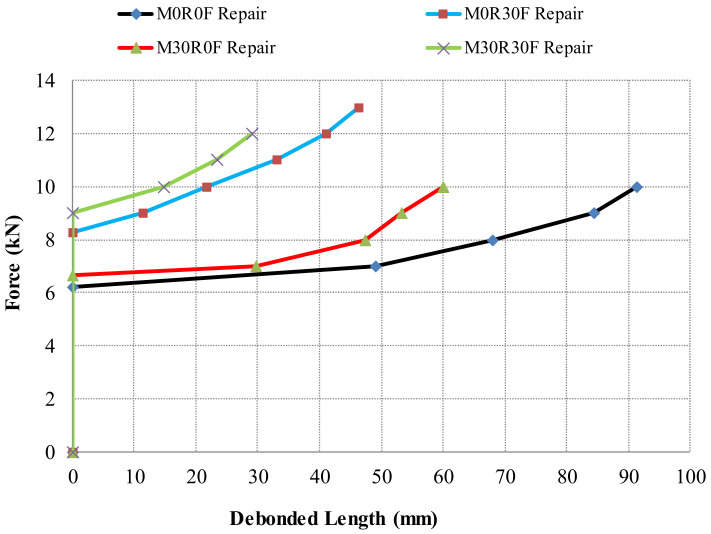
Force versus debonded length for various repair layers.

**Figure 20 materials-15-03886-f020:**
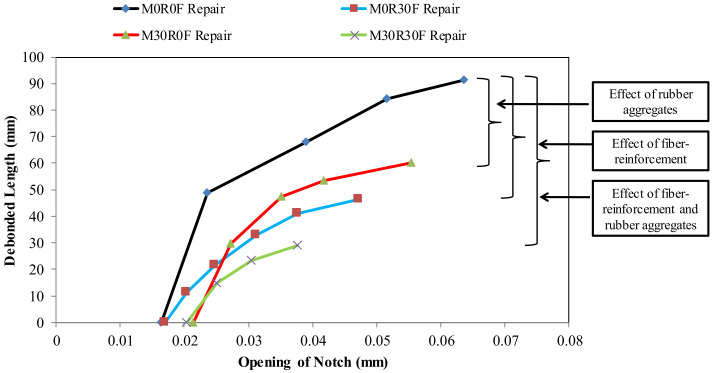
Debonded length vs. notch opening for different repair layers.

**Table 1 materials-15-03886-t001:** Physical characteristics and chemical properties of Portland cement (CEM I 52.5R).

Physical Characteristics							
Properties			Unit		Value		
Specific gravity			g/cm^3^		3.13		
Water demand			%		28.1		
Fineness			cm^2^/g		4067		
28-day compressive strength			MPa		>50		
**Chemical properties (%)**							
C_3_S + C_2_S	CaO/SiO_2_	MgO	C_3_S	C_2_S	C_3_A	C_4_AF	Gypsum
78.1	2.9	0.6	68	12	7	9	4.2

**Table 2 materials-15-03886-t002:** Properties of metallic fibres (Fibraexsaint-Gobain [21]).

Properties of Metallic Fibres	
Length, L	30 mm
Thickness	29 µm
Density	7200 kg/m^3^
Tensile strength	More than 1400 MPa
Elastic modulus	140 GPa
Raw material	Amorphous metal (Fe, Cr)80, (P, C, Si)20

**Table 3 materials-15-03886-t003:** Mixture design (kg/m^3^).

Sr. No.	Mix Designation	Cement	Rubber Aggregates	Sand	Water	Fibres	Super-Plasticizer	Viscosity Modifying Agent
1	M0R0F	500	0	1600	250	0	1.2	0
2	M0R30F					30	5	
3	M30R0F		215	1120		0	4.5	2.5
4	M30R30F					30	10	

**Table 4 materials-15-03886-t004:** Average interface debonding-initiation force.

Mix Composition	M0R0F	M0R30F	M30R0F	M30R30F
Average interface debonding-initiation force (kN)	6.5	8.0	7.0	9.0

**Table 5 materials-15-03886-t005:** Compressive strength and modulus of elasticity of various materials used for repair.

Repair Material	Compressive Strength (MPa)	Modulus of Elasticity (Mpa)
M0R0F	52.7	28,580
M0R30F	51.6	28,332
M30R0F	13.0	10,840
M30R30F	9.1	10,775

**Table 6 materials-15-03886-t006:** Experimental data for calibrated model.

Repair Material	M0R0F	M0R30F	M30R0F	M30R30F
*w_l_* (mm)	0.125	0.72	0.36	0.99
*R_t_* (MPa)	3.17	3.2	1.02	1.98

## Data Availability

Not applicable.

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
