# Peer review of "Debonding of Thin Bonded Rubberised Fibre-Reinforced Cement-Based Repairs under Monotonic Loading: Experimental and Numerical Investigation"

_materials, 2022, doi:10.3390/ma15113886_

Round 1

Reviewer 1 Report

--Review of: “Debonding of thin bonded rubberized fibre-reinforced cement-based repairs under monotonic loading: experimental and numerical investigation” by Syed Asad Ali Gillani et al. The experiments of this manuscript have been carried out well and some questions are below to help you improve the manuscript.

--Lines 44~46; ‘Concrete is beingextensively utilised in the construction sector from the past decades and withtime reduction in the load retaining capacity of existing infrastructures has been observed.’ What’re means of the ‘beingextensively’ and ‘withtime’? Please clarify.

--The research gaps and contributions of this manuscript have not been identified in the last paragraph of the Introduction Section. In the last paragraph, the authors have summarized what has been done instead of discussing why it's done. Please clarify.

--The chemical composition of the Portland cement (CEM I 52.5R) can be presented in the section ‘2. Materials’, which is the important information.

--Lines 134~135; ‘Amorphous metallicfibres were obtained from Fibraflex Saint-Gobain[19] and are shown in Figure 3.’, Figure 3 is Figure 2?

--Lines 208~209;’After casting, these substrates were cured for three months under control environment of 20 oC and 100% relative humidity (RH).’, the 100% relative humidity can be carried out in the actual experiment? Generally, the relative humidity larger than 95% is OK

--The microstructure of cement-based repairs can be investigated for the influences of fiber and rubber aggregate on the materials, which is significant for the mechanical characterization.

Author Response

Please find the attached file for point by point reply to reviewer's comments. 

Reviewer 2 Report

The paper “Materials-1719778-Debonding of thin bonded rubberised fibre-reinforced cement-based repairs under monotonic loading: Experimental and numerical investigation” investigates the structural behaviour of composite beam (base and its repair) under monotonic loading. For this purpose, three point-bending monotonic tests were performed and, crack and transmission of delamination along the interface between the substrate and repair were monitored with Digital 3D image correlation (DIC). The crack propagation in the fibre-reinforced or/and rubberised repair materials and the debonding between substrate and different repair materials were modelled, as well. Finite element method (FEM) was utilized to estimate the mechanical response of the composite beams.

The topic of the study is original. The paper is in the scope of the journal. The study is a dense experimental and numerical modelling study.

In this study, the related literature was sufficiently mentioned in the “introduction section”.

Experimental design and the followed methodology are good and satisfactory. Obtained results and mechanisms underneath the reactions/behaviours were well explained/evaluated and discussed with the literature.

This paper can significantly contribute to the civil engineers, researchers, scholars, etc.

Some other comment/s are given below.

***The difference of the study and the novelty statements were not sufficiently given.

***There are a lot of typing errors in the body of the text. A native speaker should also check the language to further increase the quality of the paper.

***Line 135: “………………shown in Figure 3. The length of the…………..” Here, “Figure 3” should be corrected as “Figure 2”.

***Font size on the figures, font size of figure titles and values on the x/y axis should be re-written with a same font size.

***“Conclusions” part is very long. Instead, please write “conclusions” part, concisely.

Author Response

Please find the attached file with point by point reply to reviewer comments. 
